# Expansion of the Analytical Modeling of Capacitance for 1-N-1 Multilayered CID Structures with Monotonically Increasing/Decreasing Permittivity

**DOI:** 10.3390/s23135838

**Published:** 2023-06-23

**Authors:** Anwar Ulla Khan

**Affiliations:** Department of Electrical Engineering Technology, College of Applied Industrial Technology, Jazan University, Jazan 45142, Saudi Arabia; anwarkhan@jazanu.edu.sa

**Keywords:** coplanar interdigitated (CID) electrodes, conformal mapping (CM), partial capacitance (PC) techniques, capacitive sensors, infectious disease management

## Abstract

Capacitive sensors that utilize the Coplanar Interdigitated (CID) electrode structure are widely employed in various technical and analytical domains, such as healthcare, infectious disease management, pharmaceuticals, metrology, and environmental monitoring. The present exigency for lab-on-a-chip contrivances and the requisite for the miniaturization of sensors have led to the widespread adoption of CID sensors featuring multiple dielectric layers (DLs), either in the form of substrates or superstrates. Previously, we derived an analytical model for the capacitance of CID capacitive sensors with four distinct 1-N-1 patterns (namely, 1-1-1, 1-3-1, 1-5-1, and 1-11-1) using partial capacitance (PC) and conformal mapping (CM) techniques. The aforementioned model has been employed in various applications wherein the permittivity of successive layers exhibits a monotonic decrease as one moves away from the electrode plane, resulting in highly satisfactory outcomes. Nevertheless, the PC technique is inadequate for structures with multiple layers where the permittivity exhibits a monotonic increase as the distance from the electrodes increases. Given these circumstances, it is necessary to adapt the initial PC method to incorporate these novel configurations. In this work, we have discussed a new approach, splitting the concept of PC into partial parallel capacitance (PPC) and partial serial capacitance (PSC), where new CM transformations are proposed for the latter case. Thus, the present study proposes a novel methodology to expand upon our prior analytical framework, which aims to incorporate scenarios where the permittivity experiences a reduction across successive layers. The outcomes are juxtaposed with the finite element simulation and analytical findings.

## 1. Introduction

Coplanar Interdigitated (CID) capacitive sensors are extensively used to calculate near-surface parameters, including conductivity, permeability, and dielectric properties. Therefore, a more accurate description of their electrical performance is required. Depending on the sensor configuration selected and the properties of the substance being tested, the conditions under which these various types of sensors can be utilized will vary. The majority of CID configurations find use in bacterial detection [1,2], in liquid detection as a noninvasive sensor in chemical and biological fields [3,4,5], in surface acoustic wave (SAW) sensors [6,7], in the detection of tainted seafood [8], in biosensor applications [9,10], and in the advancement of MEMS fabrication technologies [11]. Several attempts have been made to enhance the sensing capabilities and maximum field penetration depth (PD) and to estimate the capacitance value of the CID sensor by altering its geometrical configurations [12,13]. The study conducted by A. R. Mohd Syaifudin et al. [13] investigated the impact of the optimal quantity of negative electrodes (NEs) positioned between two positive electrodes (PEs) with a CID configuration on the measurement sensitivity. An analysis was conducted to examine the effects of different quantities of NEs situated between two positive electrodes. The authors put forth four distinct configurations of CID sensors, each utilizing a 1-N-1 structure. These configurations, namely, 1-1-1, 1-3-1, 1-5-1, and 1-11-1, were designed to possess identical effective areas, pitches, lengths, widths, and equivalent electrode numbers. All sensor configurations possess a coplanar morphology and exhibit a straightforward architecture. Closed-form expressions are necessary for accurately computing the capacitance of CID sensors with varying geometrical patterns, taking into account the chemical-sensitive layer and substrate properties.

The optimization of operating sensitivity is a crucial aspect of a miniaturized sensor. The predominant methodology employed involves the utilization of a numerical approach, such as the finite element method (FEM), to model the entire structure and evaluate the electromagnetic field’s distribution. Although the numerical method has the capability to yield precise and dependable results, the iterative procedure of altering the structure, configuring the parameters, and processing the data can be deemed as a laborious and ineffective process. In addition, a significant number of researchers are unable to access the most precise commercial numerical tools due to their high cost. An economical and effective analytical methodology that can depict the electromagnetic field distribution within the CID structure and evaluate the conduct and attributes of the CID sensor may offer significant benefits.

To date, several exemplary models have been documented for CID electrodes with multilayer structures [14,15,16,17,18,19,20,21,22]. Alley [14] proposed an estimated model for a CID capacitor utilizing a loss-less integrated microstrip line principle. This model can aid in estimating the capacitance values of CID capacitors with uniform electrode width and gap width, specifically for the uppermost infinite air layer. In their study, Esfandiari et al. [15] made modifications to Alley’s model by integrating the impact of metallization thickness on the total capacitance measurement. Wei [16] proposed a CID sensor capacitance estimation model in the situation of an infinite uppermost air layer, utilizing conformal mapping CM techniques. The model’s predictive capacity was found to be insufficient for CID structures that feature a DL of finite thickness or a configuration with multiple layers on the electrodes.

The initial proposal for a multilayered configuration was put forth by Wu et al. in their work [17]. Gevorgian et al. [18] introduced a novel multilayered top structure model for a CID electrode sensor that utilizes CM techniques. This model differs from the model presented by Wu. The Gevorgian model exhibits a notable drawback: the capacitance values estimated through its implementation do not align with those obtained through experimentation.

A novel approach was introduced in [19] that utilizes conformal transformations and the PC method [20] to estimate the total capacitance values of multilayered structures in CID sensors. The present model takes into account the fringing capacitances arising from the outer fingers. Nonetheless, the aforementioned model proved inadequate when applied to structures with multiple layers, wherein the permittivity continuously increases as one moves farther from the electrodes. In their work [21], R. Igreja et al. presented a modified model and a novel approach involving dividing PC into PPC and PSC. The authors also introduced new CM transformations for the latter case. Consequently, this innovative methodology expands upon their prior analytical framework to accommodate scenarios in which permittivity is reduced from one layer to the next.

The applicability of all proposed models is limited to the 1-1-1 CID pattern, which consists of one NE positioned between two PEs in the context of the CID sensor. Our prior work [22] proposed a model utilizing CM transformations and PC techniques to derive expressions for estimating sensor capacitances for all feasible configurations of CID sensors (1-N-1) in multilayered structures. This model incorporates considerations for the impact of fringing field capacitance resulting from the external electrodes of the CID capacitive sensor. The present study provides a comprehensive analysis of the theoretical aspects of the CID sensor, considering various geometrical configurations. The permittivity of the layers situated in the upper or lower half-plane in relation to the electrodes exhibits a decrease as the distance from the electrode plane increases. The PC is a highly effective technique that yields precise outcomes in this scenario.

In modern times, there has been a growing demand for compacting and incorporating CID electrodes on limited surface areas. Modern lab-on-a-chip (LOC) devices have been developed to meet this demand, utilizing conventional microelectronic fabrication techniques such as Si/SiO2 substrates [23]. These techniques are employed to yield highly polished surfaces for the deposition of electrodes or to apply protective passivation layers atop the electrodes [24]. In several instances, the superstrate and/or the substrate’s permittivity exhibits a non-monotonic reduction as we depart from the electrode plane. Consequently, as elaborated in the forthcoming sections, the conventional partial capacitance technique becomes inadequate in providing precise outcomes.

As previously stated, the initial PC method is inappropriate for implementation in complex structures where the permittivity continuously increases with distance from the electrodes. In the scenario described, the EF near the interface separating two contiguous dielectric layers (DLs) exhibits a perpendicular orientation with respect to the interfaces. This behavior is analogous to the presence of a Dirichlet boundary condition (DBC), in which the EF sustains an unchanging magnitude along the boundary. Consequently, as initially outlined, the PC technique is inadequate in providing precise outcomes.

Zhu et al. [25] raised concerns regarding the effectiveness of the PC technique in scenarios where the permittivity exhibits a monotonic reduction as the distance from the electrode plane increases. Specifically, the authors examined coplanar waveguides and proposed that a serial decomposition approach may be more appropriate than a parallel decomposition approach. Ghione et al. proposed a modification to the PC technique for coplanar waveguides that involves separating the situation into three distinct cases [26,27]. The first case involves a monotonically decreasing permittivity as one moves away from the plane of the electrodes, while the second case involves a monotonically increasing permittivity in the same direction. The third case is a mixed scenario where there is no discernible monotonic behavior for the permittivity. Ghione et al. demonstrated that it is feasible to assess the effectiveness of the PC method by utilizing an approximation of Green’s function. They propose that for scenario (i), the Parallel Partial Capacitance (PPC) technique should be employed, while for scenario (ii), the Serial Partial Capacitance (PSC) technique is recommended. An answer to the problem for case (iii) could not be found. Prior research [23] initially introduced the proposal in question regarding interdigital electrodes. However, the requisite conformal mapping equations were not provided at that juncture, constituting the current study’s primary objective. To partition the problem into PPC or SPC, it is imperative to generate novel expressions utilizing the CM methodology specifically for the SPC scenario to accommodate the novel boundary conditions.

In the current work, we have proposed a new CM transformation technique for partial serial capacitance (PSC) by splitting the concept of PC into partial parallel capacitance (PPC) and partial serial capacitance (PSC) to obtain an analytical expression (model) for the capacitance of CID capacitive sensors for four 1-N-1 patterns (such as 1-1-1, 1-3-1, 1-5-1, and 1-11-1) with monotonically increasing/decreasing permittivity. This model also considers the effects of the CID sensor’s outer electrodes’ capacitance-causing fringe field capacitance. A detailed study of the theory of the CID sensor with various geometrical configurations is provided. MATLAB has been utilized to analyze the multiple patterns of 1-N-1 CID sensors. The 1-N-1 CID electrode structure has been designed and simulated with finite element software in order to validate the proposed analytical model and simulation results.

## 2. Physical Model of the CID Sensor

The CID capacitive sensor employs the same operating principle as a parallel plate capacitor. The CID sensor’s electrode pattern can be repeated numerous times in order to generate a potent signal. EF distribution between PE and NE can exhibit multiple excitation patterns at varying levels of proximity for various electrode arrangements with optimal pitch lengths. Two adjacent electrodes with similar polarity can be used to calculate the CID sensor’s penetration depth (PD). Based on the information mentioned above, four distinct electrode patterns (1-1-1, 1-3-1, 1-5-1, and 1-11-1) have been devised with optimal numbers of NEs, deeper penetration, and uniform EF distribution throughout the sensor geometry [13]. The optimal number of NEs between two PEs of the CID sensor pattern contributes to the most precise sensitivity measurement. The 1-1-1 pattern sensor exhibits a high signal intensity but a relatively small PD, whereas the 1-11-1 pattern sensor depicts the opposite. When designing the sensor, a compromise must be made between the intended signal strength (in terms of equivalent capacitance) and the PD. Therefore, 1-3-1 or 1-5-1 may be the optimal choice for moderate signal intensity and depth of penetration [12].

The 1-N-1 CID capacitive sensor patterns with 13 fingers are depicted in Figure 1, illustrating the four potential patterns (1-1-1, 1-3-1, 1-5-1, and 1-11-1). The layout of the schematic diagram of the periodic Coplanar Interdigitated (CID) cross-section with multiple dielectric layers on the upper and lower half-planes is shown in Figure 2. All electrodes possess a uniform width denoted by ‘w’ and a length of ‘l.’ The distance between them is represented by ‘g.’ Each positive and negative electrode is linked to a constant voltage of +V and −V, respectively.

The determination of the total capacitance value between the PEs and NEs of all four CID patterns is contingent solely upon the two non-dimensional variables, namely, the metallization factor ‘ξ’ and the height-to-wavelength factor ‘γ’, which have been explicitly stated as
(1)$$\xi =\frac{w}{w+g}=\frac{2w}{\lambda }\;$$
and
(2)$$\gamma =\frac{h}{2\left(w+g\right)}=\frac{h}{\lambda }\;$$
where “h” represents the height of the DL (as seen from the electrode surface), and λ=2(w+g) is the electrode’s spatial wavelength (SW). The SW for the 1-N-1 (with N ≠ 1) CID pattern can be expressed as
(3)$$\lambda_{\alpha }=2\alpha \left(w+g\right)$$
where α (=1,2, ….(n+1)/2).

The analysis of Figure 3 reveals that the 1-3-1 pattern exhibits a total of two SWs, denoted as $\lambda_{1}{\;\mathrm{and}\;\lambda }_{2}$. Conversely, the 1-5-1 and 1-11-1 patterns display a more significant number of SWs, specifically three ($\lambda_{1},{\;\lambda }_{2}{\;\mathrm{and}\;\lambda }_{3})$ and six ($\lambda_{1},{\;\lambda }_{2},{\;\lambda }_{3},{\;\lambda }_{4},{\;\lambda }_{5}{\;\mathrm{and}\;\lambda }_{6})$, respectively.

The equipotential planes with a zero potential are the normal planes between the PEs and NEs of the four CID structures. This is because the EF is perpendicular to these equipotential planes, as depicted in Figure 3. The condition for a pattern to be infinitely periodic is satisfied when the Laplace equation is verified without the presence of electric charge and the length of the electrodes is significantly greater than the thickness.

In practical terms, the electrode fingers’ finite length can be deemed infinite due to their significant size in comparison to the SW of the CID sensor structure.

The electrodes’ negligible thickness relative to their width allows for considering electrode potentials between the upper and lower half-planes. Several authors have suggested incorporating electrode thickness corrections [26,28,29] to improve the accuracy of transducer measurements. However, it should be noted that while these corrections may be effective for transducers with infinite layers, they may not be suitable for multilayered structures. In instances where the thickness of the layer housing the electrodes exceeds that of the electrodes themselves, it is feasible to incorporate the impact of the parallel plate (PP) capacitor that arises between neighboring electrodes, thereby achieving precise outcomes.

Figure 3 depicts the electric circuit that corresponds to four distinct configurations (1-1-1, 1-3-1, 1-5-1, and 1-11-1) of the 1-N-1 CID pattern. These configurations consist of 13 fingers and feature a single layer above the electrode plane. Due to symmetry considerations, the total capacitance of a single layer can be assessed based on two distinct types of capacitance, as illustrated in Figure 3: (1) $C_{I\alpha }$, which is equal to half the capacitance of an interior electrode with respect to the ground voltage, and (2) $C_{E\alpha }$, which represents the capacitance between the ground and the external electrode. The variable α (where α is an integer ranging from 1 to (N+1)/2) is contingent upon the specific CID pattern being utilized. For instance, in the case of a 1-5-1 pattern, α would take on the values of 1, 2, and 3. The interior and exterior capacitances corresponding to the aforementioned entities are denoted as $C_{I1},{\;C}_{I2},{\;C}_{I3}$ and $C_{E1},{\;C}_{E2},{\;C}_{E3}$, respectively. Likewise, in the case of CID patterns: 1-1-1, 1-3-1, and 1-11-1.

By employing network analysis to assess the equivalent circuit depicted in Figure 3, it is possible to derive the comprehensive formula for the aggregate capacitance linking of the NEs and PEs of a 1-N-1 CID sensor configuration, which is equivalent to
(4)$$C_{C.I.D.,1-N-1}=\overset{\left(N+1\right)/2}{\underset{\alpha =1}{\sum }}\left[\frac{2C_{I\alpha }\times C_{E\alpha }}{C_{I\alpha }+C_{E\alpha }}+\left(k-2\right)C_{I\alpha }\right]\;$$

The variable “N” represents the number of sensing electrodes that are positioned between two consecutive Positive Electrodes (PEs). The term “k” refers to the total count of positive electrodes. It should be noted that this assertion remains valid solely under the circumstance where the initial and final electrodes are operating as PEs.

## 3. Reconsidering the Multilayer Issue

In order to solve the cumulative capacitance of a multilayered pattern, we only need to model one side of the plane (the upper side in this case) because the capacitance on the other side (lower side) can be estimated in the same manner.

As suggested by Ghione [27], we have separated the problem into two independent scenarios for each side of the plane: (1) Partial Parallel Capacitance (PPC) and (2) Partial Series Capacitance (PSC).

### 3.1. Partial Parallel Capacitance (PPC)

In this instance, the previous method of PC can be implemented, and the theory of an NB condition (magnetic wall with d∅/dn=0) at the boundary of successive layers is applicable. In order to differentiate, this technique may be referred to as Partial Parallel Capacitance (PPC), as suggested by Ghione et al. [27]. In the situation of multilayered CID structures, where the dielectric permittivity reduces monotonically as we progress farther from the electrode plane, the capacitances earlier specified as $C_{I\alpha }$ and $C_{E\alpha }$ (see Figure 2) are now parallel to the electrode plane.

$C_{I\alpha }$ and $C_{E\alpha }$ are subsequently determined based on the geometric capacitances (GCs) $C_{\mathrm{Ip}\alpha }$ and $C_{\mathrm{Ep}\alpha }$ (The first index denotes the classification of a finger as either an inner or outer finger. The second index pertains to the method of application, which is parallel in this particular case. The third index provides information regarding the configuration of the CID structure) (see Figure 4):(5)$$C_{I\alpha }=\overset{n-1}{\underset{i=1}{\sum }}(\varepsilon_{\mathrm{id}}-\varepsilon_{\left(i+1\right)d})C_{\mathrm{Ip}\alpha }\left(h_{i}\right)+\varepsilon_{\mathrm{nd}}C_{\mathrm{Ip}\alpha }\left(\infty \right)$$
(6)$$C_{E\alpha }=\overset{n-1}{\underset{i=1}{\sum }}(\varepsilon_{\mathrm{id}}-\varepsilon_{\left(i+1\right)d})C_{\mathrm{Ep}\alpha }\left(h_{i}\right)+\varepsilon_{\mathrm{nd}}C_{\mathrm{Ep}\alpha }\left(\infty \right)$$
where $\varepsilon_{\mathrm{id}}$ represents the relative permittivity of the ith layer. $C_{\mathrm{Ip}\alpha }\left(h_{i}\right)$ and $C_{\mathrm{Ep}\alpha }\left(h_{i}\right)$ represent the GC of the ith layer, assuming an NB between the ith layer and (i+1)th layer for the inner and outer fingers, respectively. The term ‘hi’ represents the layer’s height (relative to the electrode plane). $\varepsilon_{\mathrm{nd}}$ denotes the relative permittivity of the outermost layer. The capacitances with an infinite height layer (the outermost layer) are $C_{\mathrm{Ip}\alpha }\left(\infty \right)$ and $C_{\mathrm{Ep}\alpha }\left(\infty \right)$. It is essential to point out that for an infinite layer, $C_{\mathrm{Ip}\alpha }\left(\infty \right)$ = $C_{\mathrm{Is}\alpha }\left(\infty \right)$ = $C_{I\alpha }\left(\infty \right)$ and $C_{\mathrm{Ep}\alpha }\left(\infty \right)$ = $C_{\mathrm{Es}\alpha }\left(\infty \right)$ = $C_{E\alpha }\left(\infty \right),$ where $C_{\mathrm{Is}\alpha }\left(\infty \right)$ and $C_{\mathrm{Es}\alpha }\left(\infty \right)$ are the capacitances with an infinite height layer (the top layer) in the case of PSC (discussed in the next section). Applying the above Equation (4) calculates the CID sensor’s total capacitance. The computation of $C_{\mathrm{Ip}\alpha }\left(h_{i}\right)$ and $C_{\mathrm{Ep}\alpha }\left(h_{i}\right),$ was already proposed in our previous work (refer [22] for details) and is summarized in Table 1.

### 3.2. Partial Series Capacitance (PSC)

In cases where the permittivity exhibits a monotonic increase and the EF is predominantly oriented away from the electrode plane, the layers are assumed to be serially interconnected, leading to the adoption of the PSC technique. This method is known as Partial Series Capacitance (PSC), as proposed by Ghione et al. [27]. The capacitances, previously labeled $C_{I\alpha }$ and $C_{E\alpha }$, are now measured with the aid of all of the various layers above the plane of the electrode in series and can be computed in terms of the GCs $C_{\mathrm{Is}\alpha }$ and $C_{\mathrm{Es}\alpha }$ (refer Figure 5). The sum of all n layers’ contributions should now be calculated as follows [22]:(7)$$\frac{1}{C_{\mathrm{Is}\alpha }}=\overset{n-1}{\underset{i=1}{\sum }}\left(\frac{1}{\varepsilon_{\mathrm{id}}}-\frac{1}{\varepsilon_{\left(i+1\right)d}}\right)\frac{1}{C_{\mathrm{Is}\alpha }\left(h_{i}\right)}+\frac{1}{\varepsilon_{\mathrm{nd}}}\frac{1}{C_{\mathrm{Is}\alpha }\left(\infty \right)}$$
(8)$$\frac{1}{C_{\mathrm{Es}\alpha }}=\overset{n-1}{\underset{i=1}{\sum }}\left(\frac{1}{\varepsilon_{\mathrm{id}}}-\frac{1}{\varepsilon_{\left(i+1\right)d}}\right)\frac{1}{C_{\mathrm{Es}\alpha }\left(h_{i}\right)}+\frac{1}{\varepsilon_{\mathrm{nd}}}\frac{1}{C_{\mathrm{Es}\alpha }\left(\infty \right)}$$

$C_{\mathrm{Is}\alpha }\left(h_{i}\right)$ and $C_{\mathrm{Es}\alpha }\left(h_{i}\right)$ represent the GC of the ith layer, considering a DB between the ith layer and (i+1)th layer for the interior and exterior fingers, respectively. In the case of PSC, $C_{\mathrm{Is}\alpha }\left(\infty \right)$ and $C_{\mathrm{Es}\alpha }\left(\infty \right)$ are the capacitances with an infinite height layer (the top layer). Equation (4) represents the overall capacitance of the half-plane. In this instance, the GCs $C_{\mathrm{Is}\alpha }\left(h_{i}\right)$, and $C_{\mathrm{Es}\alpha }\left(h_{i}\right)$ must be determined in order to use the PSC method, as a DB condition must be investigated.

## 4. Estimating the Geometric Capacitance (GC):

### 4.1. Interior GC $C_{Is\alpha }\left(h_{i}\right)$ Estimation

The appropriate space region of the 1-N-1 CID electrode pattern will be mapped onto a PP capacitor geometry to calculate $C_{\mathrm{Is}\alpha }\left(h_{i}\right)$ using conformal mapping techniques. The present scenario involves a limited layer that exhibits a Dirichlet boundary condition, where the value of the function is zero, positioned between two neighboring layers. The representation of the x-plane on the Argand complex plane is analogous to the scenario of the NB condition between successive DLs. However, the ground electrode has been extended to the upper boundary (i.e., the boundary between two successive DLs), as depicted in Figure 6. Four conformal transformations are employed, which are depicted on complex Argand planes (refer to Figure 6). The variables x, z, t, y, and w denote complex values.

The initial x-plane is transformed onto the complex z-plane while maintaining its original aspect ratio, utilizing the complete elliptic integral of the first kind.
(9)$$z=\frac{4K\left(k_{\mathrm{Is}\alpha }\right)}{\lambda_{s\alpha }}x\;$$

The variable $K\left(k_{\mathrm{Is}\alpha }\right)$ represents the complete elliptic integral of the 1st kind with modulus $k_{\mathrm{Is}\alpha }$.
(10)$$k_{\mathrm{Is}\alpha }={\left(\;\frac{\upsilon_{2}\left(0,{\;Q}_{s\alpha }\right)}{\upsilon_{3}\left(0,{\;Q}_{s\alpha }\right)}\;\right)}^{2}$$

The functions $\upsilon_{2}\left(0,{\;Q}_{s\alpha }\right)$ and $\upsilon_{3}\left(0,{\;Q}_{s\alpha }\right)$ correspond to the 2nd and 3rd Jacobi functions, respectively [30],


and

(11)
$$Q_{s\alpha }=\exp\left(-4{\pi r}_{s\alpha }\right)\;$$



The rectangular shape located in the z-plane has been subjected to a mapping process onto the t-plane by utilizing a particular function.
(12)$$t_{2,\;\mathrm{Is}\alpha }=\mathrm{sn}\left[Z_{s\alpha },{\;k}_{s\alpha }\right]\;$$

The function $\mathrm{sn}\left[Z_{s\alpha },{\;k}_{s\alpha }\right]$ denotes the Jacobi elliptic function with a modulus of $k_{s\alpha }$.

Subsequently, the t-plane undergoes a mapping process that results in its transformation onto the y-plane, which can be expressed as
(13)$$y=\frac{t}{t_{2,\;\mathrm{Is}\alpha }}\;$$

Ultimately, the y-plane undergoes a transformation into the w-plane through the utilization of the SC transformation. Specifically, the upper semi-plane within the y-plane is mapped onto the interior of the rectangle located within the w-plane.
(14)$$w=F\left(\emptyset ,{\;k}_{\mathrm{Is}\alpha }\right)\;$$
where
$$k_{\mathrm{Is}\alpha }=t_{2,\;\mathrm{Is}\alpha }\;\mathrm{and}\;\emptyset =\sin^{-1}y$$

Furthermore, the symbol $F\left(\emptyset ,{\;k}_{\mathrm{Is}\alpha }\right)$ represents the first kind of incomplete elliptic integral with modulus.

A PP electrode is generated upon performing the electrode transformation from the x-plane to the w-plane, which constitutes the principal characteristic of this particular series of transformations. Therefore, given the dimensions of the PP capacitor in the w-plane, it is possible to readily approximate the capacitance value in the x-plane using the aforementioned formula.
(15)$$C_{\mathrm{Is}\alpha }\left(h_{i}\right)=\varepsilon_{o}L\left[\frac{K\left(k_{\mathrm{Is}\alpha }\right)}{K\left({\;k^{\prime }}_{\mathrm{Is}\alpha }\right)}\right]$$
where
(16)$${k^{\prime }}_{\mathrm{Is}\alpha }=\sqrt{1-k_{\mathrm{Is}\alpha }^{2}}\;$$

It can be observed that the variable $C_{\mathrm{Is}\alpha }\left(h_{i}\right)/L$ is solely contingent upon the two non-dimensional parameters, namely, $\eta_{s\alpha }$ and $r_{s\alpha }$.

### 4.2. Exterior GC $C_{Es\alpha }\left(h_{i}\right)$ Estimation

The computation of variable ${\;C}_{\mathrm{Es}\alpha }\left(h_{i}\right)$ involves the depiction of the initial physical domain on the Argand complex x-plane, as illustrated in Figure 7. The ground line has been extended to the boundary that separates two contiguous layers ($x_{1}=\mathrm{jh})$. The transformation utilized to map the semi-infinite strip on the x-plane onto the upper t-plane is as follows:(17)$$t=\cosh\left(\frac{\pi }{2h_{i}}x\right)\;$$

Subsequently, the t-plane undergoes a mapping process onto the y-plane through the utilization of a mapping function:(18)$$y=t\sqrt{\frac{t_{4,\;\mathrm{Is}\alpha }^{2}-\;t}{t_{4,\;\mathrm{Is}\alpha }^{2}-{\;t}_{3}}\;}$$

The y-plane has been shifted onto the w-plane through the utilization of the SC transformation [31].
(19)$$w=\int_{0}^{y}\frac{{\mathrm{dw}}^{\prime }}{\sqrt{\left(1-{w^{\prime }}^{2}\right)\left(1-k_{\mathrm{Es}\alpha }{}^{2}{w^{\prime }}^{2}\right)}}\;$$
where
(20)$$k_{\mathrm{Es}\alpha }=\sqrt{\frac{t_{4,\;\mathrm{Es}\alpha }^{2}-t_{3,\;\mathrm{Es}\alpha }^{2}}{t_{4,\;\mathrm{Es}\alpha }^{2}-1}}$$
where $t_{3,\;\mathrm{Es}\alpha }=\cosh\left(\frac{\pi \left(1-\eta_{\alpha }\right)}{4r_{\alpha }}\right)$ and $t_{4,\;\mathrm{Es}\alpha }=\cosh\left(\frac{\pi \left(1+\eta_{\alpha }\right)}{4r_{\alpha }}\right)$
(21)$${k^{\prime }}_{\mathrm{Es}\alpha }=\sqrt{1-k_{\mathrm{Es}\alpha }^{2}}$$

The shifting of electrodes from the x-plane to the w-plane results in the acquisition of a PP.

Therefore, if one possesses knowledge of the dimensions of the PP capacitor after transformation in the w-plane, it is feasible to effortlessly approximate the capacitance in the x-plane using the following method:(22)$$C_{\mathrm{Es}\alpha }\left(h_{i}\right)=\varepsilon_{\mathrm{id}}L\left[\frac{K\left(k_{\mathrm{Es}\alpha }\right)}{K\left({k^{\prime }}_{\mathrm{Es}\alpha }\right)}\right]\;$$

### 4.3. Interior GC for an Infinite Layer $C_{E\alpha }\left(\infty \right)$ Estimation

The transformations required for the calculation of $C_{I\alpha }\left(\infty \right)$ are as depicted in Figure 8.
(23)$$t=\frac{1}{k_{\mathrm{Is}\alpha \infty }}\sin\left(\frac{2\pi }{\lambda }z\right)$$
and
(24)$$w=\int_{0}^{t}\frac{{\mathrm{dw}}^{\prime }}{\sqrt{\left(1-{w^{\prime }}^{2}\right)\left(1-k_{\mathrm{Is}\alpha \infty }{}^{2}{w^{\prime }}^{2}\right)}}$$ with

(25)$$k_{\mathrm{Is}\alpha \infty }=\sin\left(\frac{\pi }{2}\eta_{s\alpha }\right)\;$$
and
(26)$${k^{\prime }}_{\mathrm{Is}\alpha \infty }=\sqrt{1-k_{\mathrm{Is}\alpha \infty }^{2}}$$

Therefore, the value of capacitance can be expressed as
(27)$$C_{I\alpha }\left(\infty \right)=\varepsilon_{o}L\left[\frac{K\left(k_{\mathrm{Is}\alpha \infty }\right)}{K\left({k^{\prime }}_{\mathrm{Is}\alpha \infty }\right)}\right]\;$$

### 4.4. Exterior GC for an Infinite Layer $C_{E\alpha }\left(\infty \right)$ Estimation

The expression of $C_{E\alpha }\left(\infty \right)$ depicted in Figure 9 was obtained through the use of transformations, as follows:

(28)$$t=\frac{2}{g_{p\alpha }}z\;$$
and
(29)$$w=\int_{0}^{t}\frac{{\mathrm{dw}}^{\prime }}{\sqrt{\left(1-{\;w^{\prime }}^{2}\right)\left(1-k_{p\alpha \infty }{}^{2}{\;w^{\prime }}^{2}\right)}}\;$$
where
(30)$$k_{p\alpha \infty }=\frac{1-\eta_{s\alpha }}{1+\eta_{s\alpha }}\;$$
(31)$$k_{\mathrm{Es}\alpha \infty }=\sqrt{1-k_{s\alpha \infty }{}^{2}\;}$$
(32)$$k_{\mathrm{Es}\alpha \infty }=\frac{2\sqrt{\eta_{s\alpha }}}{1+\eta_{s\alpha }}$$
and
(33)$${k^{\prime }}_{\mathrm{Es}\alpha \infty }=\sqrt{1-k_{\mathrm{Es}\alpha \infty }^{2}}\;$$

Ultimately, the resulting capacitance is as follows:(34)$$C_{\mathrm{Ep}\alpha }\left(\infty \right)=\varepsilon_{o}L\left[\frac{K\left(k_{\mathrm{Es}\alpha \infty }\right)}{K\left({k^{\prime }}_{\mathrm{Es}\alpha \infty }\right)}\right]$$

Table 2 and Table 3 provide a detailed explanation of the important expressions utilized in the computation of capacitances:${\;C}_{\mathrm{Is}\alpha }\left(h_{i}\right),{\;C}_{\mathrm{Es}\alpha }\left(h_{i}\right),{\;C}_{I\alpha }\left(\infty \right),\;{\mathrm{and}\;C}_{E\alpha }\left(\infty \right)$.

## 5. Results and Discussion

This section presents a comparison between the outcomes derived from the analytical model of PPC and PSC methodology and the two-dimensional finite element methods (FEMs) produced by COMSOL Multiphysics. The scope of the models is confined to uncomplicated CID electrode configurations featuring four unique 1-N-1 patterns (namely, 1-1-1, 1-3-1, 1-5-1, and 1-11-1), which yield a comprehensive two-dimensional cross-sectional representation. It is not easy to model structures of greater complexity in the horizontal plane. Nevertheless, these uncomplicated configurations are frequently encountered in research papers. Additionally, the electrode fingers must possess adequate length to disregard fringing field effects in proximity to the ends of each electrode finger. According to the observations presented in reference [32], in the case of two-electrode structures, it is recommended that the finger length L be approximately ten times greater than λ to avoid significant errors. Further, the thickness of the electrode fingers is not taken into account. This assumption may not be suitable when the thickness is comparable to the lateral dimensions of the electrode, specifically w and g. The significance of these uncomplicated models lies in their minimal computational cost compared to numerical simulations while still providing adequate precision as preliminary estimators for the capacitance of CID structures.

Figure 10a–d show the values of total capacitance per unit length $C_{C.I.D.,1-N-1}/L$ as a function of the ratio between the relative permittivity of the layers (i.e., ε_1d_/ε_2d_) for γ=0.3 (as an example) and ξ=0.5 for all possible distinct 1-N-1 patterns (1-1-1, 1-3-1, 1-5-1, and 1-11-1). Notably, the dependence of the total capacitance is not on SW (λ) but is instead on the dimensional parameters ξ and γ.

The continuous line was derived using the equations formulated in the outcomes of the PPC analytical study. In contrast, the dotted line was derived using the equations developed in the results of the PSC analytical study (refer to Table I, II, and III).

The triangular symbols in Figure 10 represent numerical values obtained from FEM simulations. The findings indicate that, while the previous model demonstrated a strong correlation between the FEM values and the continuous line of the parallel partial curve (PPC), this relationship was only observed when the ratio of ε_1d_/ε_2d_ exceeded 1. However, in instances where the ratio of ε_1d_/ε_2d_ was less than 1, the PPC approach was unable to yield precise outcomes. This suggests limitations in the applicability of the PPC method under certain conditions. When the ratio of ε_1d_ to ε_2d_ is less than 1, a complete correspondence is noted between the PSC curve (represented by a dotted line) and the FEM analysis. Various simulations have been conducted, utilizing different values of γ, from 0.05 to 0.5, for PPC and PSC with all four CID patterns (1-1-1, 1-3-1, 1-5-1, and 1-11-1). In the worst-case scenario, the maximum error was determined to be approximately 4%. Nevertheless, it was observed that the approximations for PPC and PSC were highly precise when the ratio of ε_1d_/ε_2d_ was significantly greater than 1 (in the case of PPC) and when the ratio of ε_1d_/ε_2d_ was considerably less than 1 (in the case of PSC), or when ε_1d_/ε_2d_ approached 1, in both cases.

The findings are consistent with the conclusions outlined in reference [27], which compares the PSC and PPC approaches to the spectral-domain static Green’s function method. The results depicted in Figure 10a through Figure 10d demonstrate that the utilization of both PPC and PSC techniques in conjunction is necessary to achieve precise outcomes across the complete spectrum of relative permittivity values for both layers in all four unique 1-N-1 patterns (1-1-1, 1-3-1, 1-5-1, and 1-11-1).

## 6. Conclusions

This article has successfully employed a new CM technique to make comparatively simple expressions (model) for the capacitance estimation of different 1-N-1 multilayered CID patterns with monotonically increasing/decreasing permittivity. In order to achieve this objective, the PC technique was bifurcated into PPC and PSC, and novel analytical expressions were suggested for the PSC scenario utilizing the principles of conformal mapping (CM). Thus, enhanced model robustness resulted in precise outcomes for the commonly utilized structures featuring CID electrodes. The results acquired through this expanded model exhibit satisfactory concurrence with the FEM simulations. Subsequent research endeavors will involve establishing a generalization of this developed model for multiple layers that lack a monotonic behavior of the permittivity among neighboring layers. This can be achieved through the utilization of a mixed PC decomposition.

## Figures and Tables

**Figure 1 sensors-23-05838-f001:**
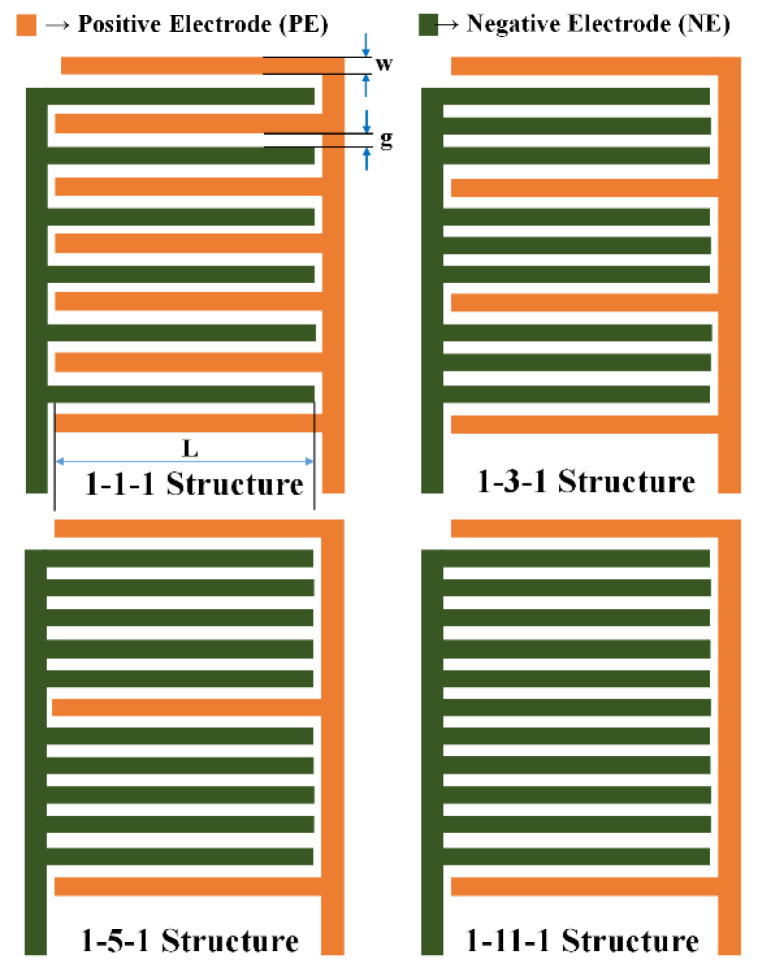
The diagram illustrating the four plausible configurations of the CID electrode structure.

**Figure 2 sensors-23-05838-f002:**
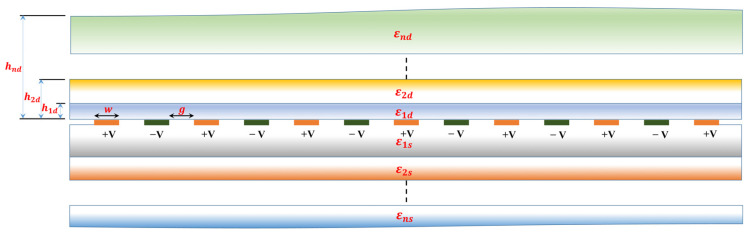
The periodic Coplanar Interdigitated (CID) cross-section with multiple dielectric layers on upper and lower half-planes.

**Figure 3 sensors-23-05838-f003:**
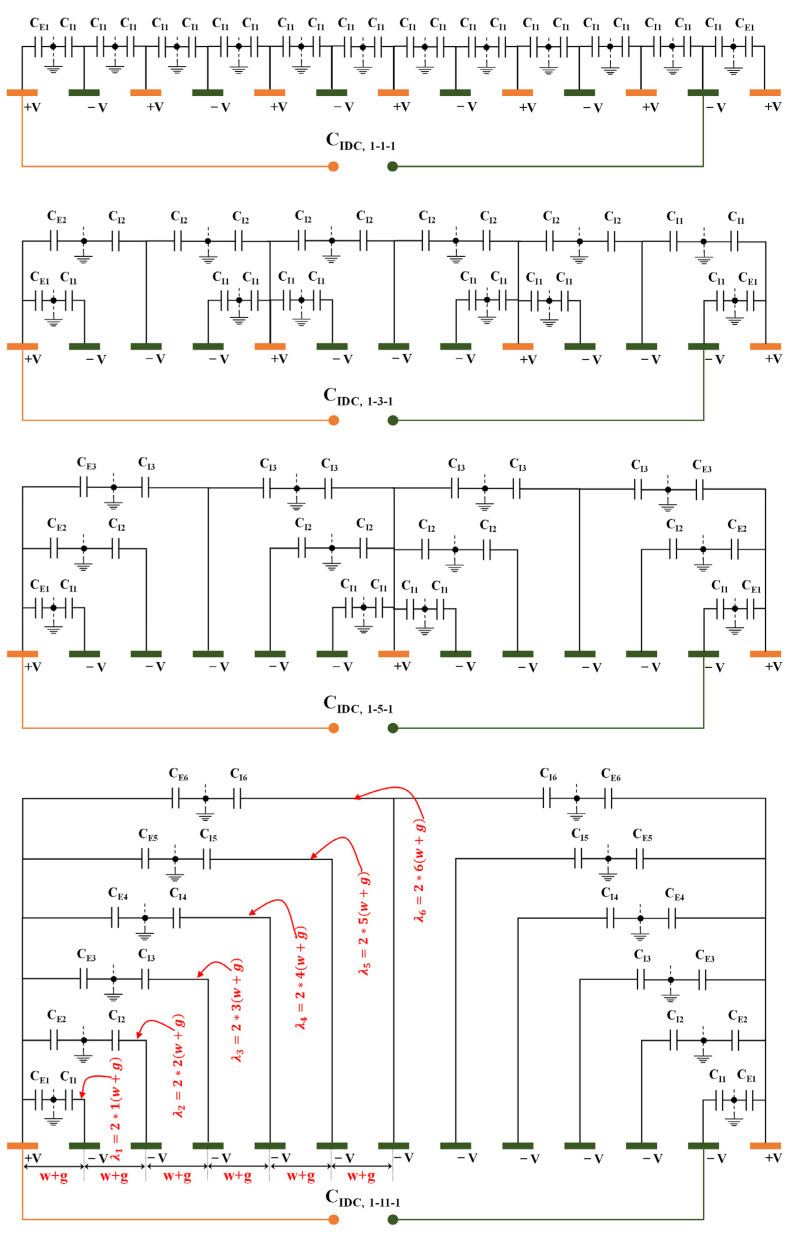
A comprehensive analysis of the various patterns that can be utilized to determine the total capacitance of a semi-infinite top layer featuring 13 fingers.

**Figure 4 sensors-23-05838-f004:**
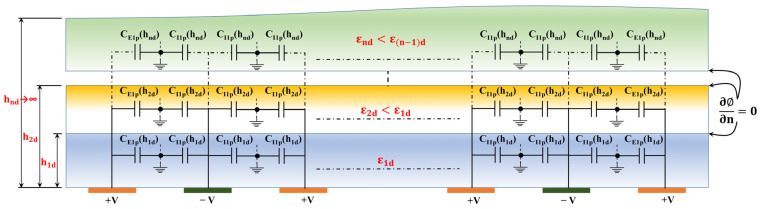
PPC technique used to determine capacitance at the interfaces between two successive layers under an NB condition. The measurement of the layer height is conducted with reference to the plane of the electrodes.

**Figure 5 sensors-23-05838-f005:**
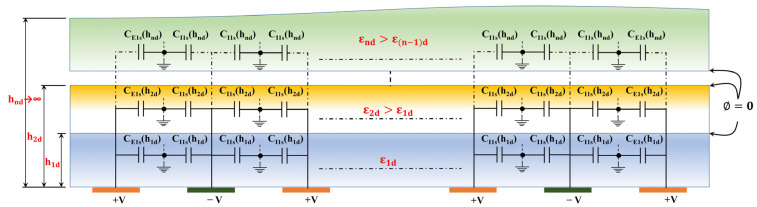
PSC technique used to determine capacitance at the interfaces between two successive layers under a DB condition. The measurement of the layer height is conducted with reference to the plane of the electrodes.

**Figure 6 sensors-23-05838-f006:**
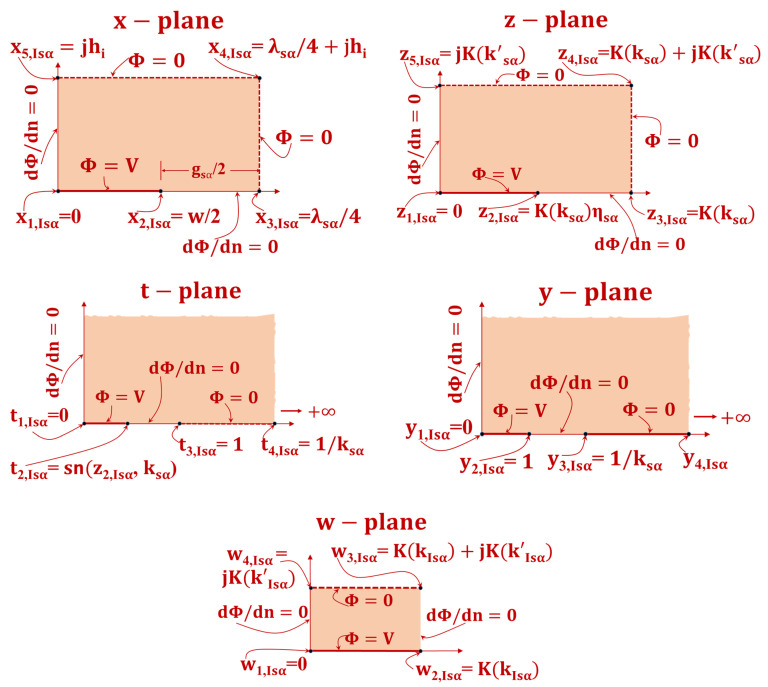
Conformal transformation methods for calculating $C_{\mathrm{Is}\alpha }\left(h_{i}\right)$. The solid line shows the equipotential line transition, while the shaded zone shows the dielectric transformation.

**Figure 7 sensors-23-05838-f007:**
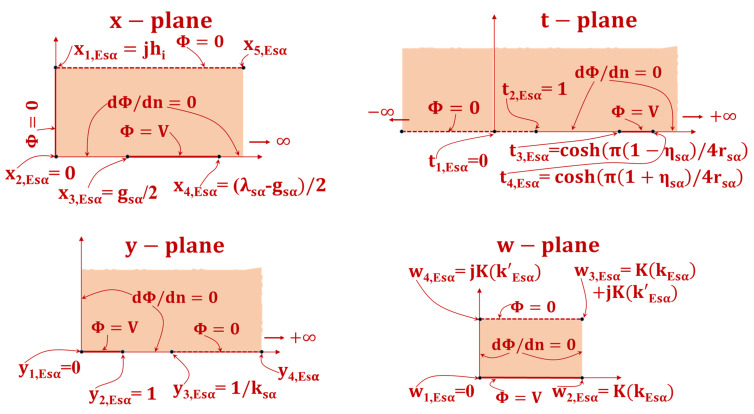
Conformal transformation methods for calculating $C_{\mathrm{Es}\alpha }\left(h_{i}\right)$. The solid line shows the equipotential line transition, while the shaded zone shows the dielectric transformation.

**Figure 8 sensors-23-05838-f008:**
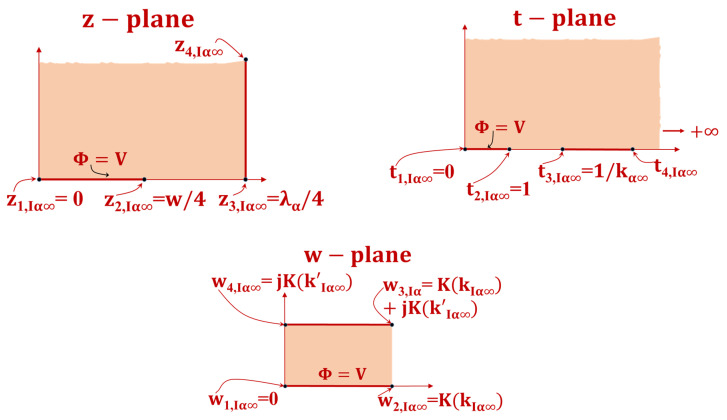
Conformal transformation methods for calculating $C_{I\alpha }\left(\infty \right)\;$. The solid line shows the equipotential line transition, while the shaded zone shows the dielectric transformation.

**Figure 9 sensors-23-05838-f009:**
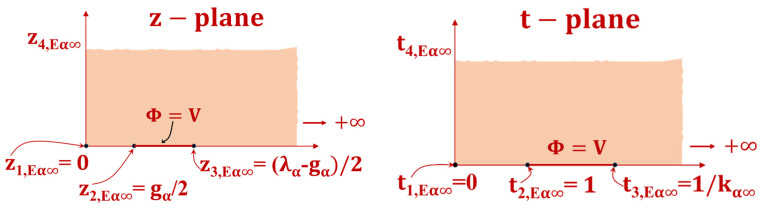
Conformal transformation methods for calculating $C_{E\alpha }\left(\infty \right)\;$. The solid line shows the equipotential line transition, while the shaded zone shows the dielectric transformation.

**Figure 10 sensors-23-05838-f010:**
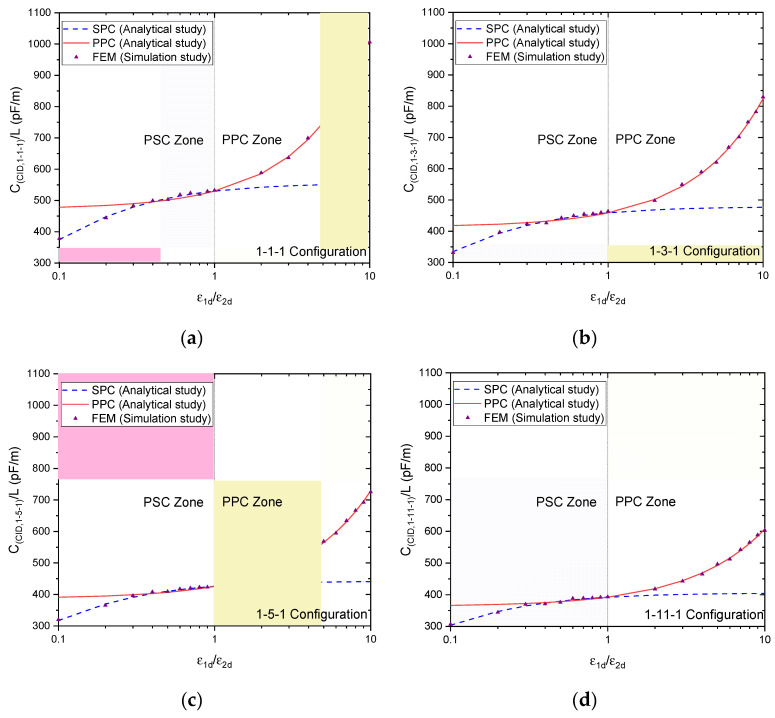
Total capacitance per unit length for the upper half-plane with two DLs as a function of the permittivity ratio between the layers for all patterns: (**a**) 1-1-1, (**b**) 1-3-1, (**c**) 1-5-1, and (**d**) 1-11-1, comparison between the models suggested, and simulation results.

**Table 1 sensors-23-05838-t001:** Important equations required for the calculation of $C_{\mathrm{Ip}\alpha }\left(h_{i}\right)\;\mathrm{and}\;C_{\mathrm{Ep}\alpha }\left(h_{i}\right)$ for PSC.

	Interior Electrodes	Exterior Electrodes
Finite Layer	$C_{\mathrm{Ip}\alpha }\left(h_{i}\right)=\varepsilon_{o}L\left[\frac{K\left(k_{\mathrm{Ip}\alpha }\right)}{K\left({\;k^{\prime }}_{\mathrm{Ip}\alpha }\right)}\right]$	$C_{\mathrm{Ep}\alpha }\left(h_{i}\right)=\varepsilon_{d}L\left[\frac{K\left(k_{\mathrm{Ep}\alpha }\right)}{K\left({k^{\prime }}_{\mathrm{Ep}\alpha }\right)}\right]$
	${k^{\prime }}_{\mathrm{Ip}\alpha }=\sqrt{1-k_{\mathrm{Ip}\alpha }^{2}}$	${k^{\prime }}_{\mathrm{Ep}\alpha }=\sqrt{1-k_{\mathrm{Ep}\alpha }^{2}}$
	$k_{\mathrm{Ip}\alpha }=t_{2,\;\mathrm{Ip}\alpha }\sqrt{\frac{t_{4,\;\mathrm{Ip}\alpha }^{2}-1}{t_{4,\;\mathrm{Ip}\alpha }^{2}-t_{2,\;\mathrm{Ip}\alpha }^{2}}\;}$	$k_{\mathrm{Ep}\alpha }=\frac{1}{t_{3,\;\mathrm{Ep}\alpha }}\sqrt{\frac{t_{4,\;\mathrm{Ep}\alpha }^{2}-t_{3,\;\mathrm{Ep}\alpha }^{2}}{t_{4,\;\mathrm{Ep}\alpha }^{2}-1}}$
	$t_{4,\;\mathrm{Ip}\alpha }=\frac{1}{k_{p\alpha }}$	$t_{3,\;\mathrm{Ep}\alpha }=\cosh\left(\frac{\pi \left(1-\eta_{p\alpha }\right)}{8r_{p\alpha }}\right)$
	$t_{2,\;\mathrm{Ip}\alpha }=\mathrm{sn}\left[K\left(k_{p\alpha }\right)\eta_{p\alpha },{\;k}_{p\alpha }\right]$	$t_{4,\;\mathrm{Ep}\alpha }=\cosh\left(\frac{\pi \left(1+\eta_{p\alpha }\right)}{8r_{p\alpha }}\right)$
	$k_{p\alpha }={\left(\;\frac{\upsilon_{2}\left(0,{\;Q}_{p\alpha }\right)}{\upsilon_{3}\left(0,{\;Q}_{p\alpha }\right)}\;\right)}^{2}$	
	$Q_{p\alpha }=\exp\left(-4{\pi r}_{p\alpha }\right)$	

**Table 2 sensors-23-05838-t002:** Important equations required for the calculation of $C_{\mathrm{Is}\alpha }\left(h_{i}\right)\;\mathrm{and}\;C_{\mathrm{Es}\alpha }\left(h_{i}\right)$ for PSC.

	Interior Electrodes	Exterior Electrodes
Finite Layer	$C_{\mathrm{Is}\alpha }\left(h_{i}\right)=\varepsilon_{o}L\left[\frac{K\left(k_{\mathrm{Is}\alpha }\right)}{K\left({\;k^{\prime }}_{\mathrm{Is}\alpha }\right)}\right]$	$C_{\mathrm{Es}\alpha }\left(h_{i}\right)=\varepsilon_{d}L\left[\frac{K\left(k_{\mathrm{Es}\alpha }\right)}{K\left({k^{\prime }}_{\mathrm{Es}\alpha }\right)}\right]$
	${k^{\prime }}_{\mathrm{Is}\alpha }=\sqrt{1-k_{\mathrm{Is}\alpha }^{2}}$	${k^{\prime }}_{\mathrm{Es}\alpha }=\sqrt{1-k_{\mathrm{Es}\alpha }^{2}}$
	$k_{\mathrm{Is}\alpha }=t_{2,\;\mathrm{Is}\alpha }$	$k_{\mathrm{Es}\alpha }=\sqrt{\frac{t_{4,\;\mathrm{Es}\alpha }-t_{3,\;\mathrm{Es}\alpha }}{t_{4,\;\mathrm{Es}\alpha }-1}}$
	$t_{2,\;\mathrm{Is}\alpha }=\mathrm{sn}\left[K\left(k_{s\alpha }\right)\eta_{s\alpha },{\;k}_{s\alpha }\right]$	$t_{3,\;\mathrm{Es}\alpha }=\cosh\left(\frac{\pi \left(1-\eta_{s\alpha }\right)}{4r_{s\alpha }}\right)$
	-	$t_{4,\;\mathrm{Es}\alpha }=\cosh\left(\frac{\pi \left(1+\eta_{s\alpha }\right)}{4r_{s\alpha }}\right)$
	$k_{s\alpha }={\left(\;\frac{\upsilon_{2}\left(0,{\;Q}_{s\alpha }\right)}{\upsilon_{3}\left(0,{\;Q}_{s\alpha }\right)}\;\right)}^{2}$	
	$Q_{s\alpha }=\exp\left(-4{\pi r}_{s\alpha }\right)$	

**Table 3 sensors-23-05838-t003:** Important equations required for the calculation of $C_{I\alpha }\left(\infty \right),{\;\mathrm{and}\;C}_{E\alpha }\left(\infty \right)$.

	Interior Electrodes	Exterior Electrodes
Infinite layer	$C_{I\alpha }\left(\infty \right)=\varepsilon_{o}L\left[\frac{K\left(k_{I\alpha \infty }\right)}{K\left({\;k^{\prime }}_{I\alpha \infty }\right)}\right]$	$C_{E\alpha }\left(\infty \right)=\varepsilon_{o}L\left[\frac{K\left(k_{E\alpha \infty }\right)}{K\left({k^{\prime }}_{E\alpha \infty }\right)}\right]$
	${k^{\prime }}_{I\alpha \infty }=\sqrt{1-k_{I\alpha \infty }^{2}}$	${k^{\prime }}_{E\alpha \infty }=\sqrt{1-k_{E\alpha \infty }^{2}}$
	$k_{I\alpha \infty }=\sin\left(\frac{\pi }{2}\eta_{\alpha }\right)$	$k_{E\alpha \infty }=\frac{2\sqrt{\eta_{\alpha }}}{1+\eta_{\alpha }}$

## Data Availability

Not applicable.

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
