# Peer review of "Expansion of the Analytical Modeling of Capacitance for 1-N-1 Multilayered CID Structures with Monotonically Increasing/Decreasing Permittivity"

_sensors, 2023, doi:10.3390/s23135838_

Round 1

Reviewer 1 Report

This manuscript studied a model of capacitance for 1-N-1multilayered CID structures with monotonically increasing/decreasing permittivity. However, some parts of the manuscript should be modified before publication.

1.       In the last paragraph of the introduction section, the main idea of this manuscript should be introduced instead of just discussing other papers’ results.

2.       This is a research paper’s manuscript not a review manuscript. Therefore, in the first sentence of the conclusion section, the conclusion of this work should be emphasized, instead of emphasizing reviewing other publications.

The English language should be double checked.

Author Response

We are very thankful to the reviewer for their valuable comments and support in improving the quality of the paper.

We have revised the manuscript along the lines suggested by the reviewers. Detailed answers to the questions by the Reviewers and descriptions of how we considered the reviewers' comments in preparing the revised manuscript are attached below.

Reviewer 2 Report

1. The abstract should be shorten to show the overall contents and main contributions of the paper.

2. What is the main contributions of this research? In the Introduction, It is hard to find the main advantages of the proposed research compared with the existing research.

3. Throughout the paper, the authors should more emphasize the key results. Please shorten the main contents for better understanding.

The paper is well written with sufficient readability and English quality. Please correct some typos for future publication.

Author Response

(The authors gave the same response as above.)

Reviewer 3 Report

In this work, the authors provide a new approach to expend the prior analytical framework. This work presents some new insight in this field, I suggest accepting this work after the author carefully solve the following issues.

1. the author should add more discussion of the four different structures including (1-1-1, 1-3-1, 1-5-1, and 1-11-1 in manuscript. 

2. Which structure is promising in the near future, please provide some comments.

Author Response

We are very thankful to the reviewer for their valuable comments and support in improving the quality of the paper.

We have revised the manuscript along the lines suggested by the reviewers. Detailed answers to the questions by the Reviewers and descriptions of how we considered the reviewers' comments in preparing the revised manuscript are atached below.

Round 2

Reviewer 1 Report

Recommend for publication with current version. 

Reviewer 2 Report

The paper has been properly revised according to the suggested comments, and it can be accepted for future publications.